# Applications of Smart Technology as a Sustainable Strategy in Modern Swine Farming

Shad Mahfuz [1,2,†], Hong-Seok Mun [1,3,†], Muhammad Ammar Dilawar [1,4] and Chul-Ju Yang [1,4,*]

1   Animal Nutrition and Feed Science Laboratory, Department of Animal Science and Technology, Sunchon National University, Suncheon 57922, Korea; shadmahfuz@sau.ac.bd (S.M.); mhs8828@scnu.ac.kr (H.-S.M.); ammar_dilawar@yahoo.com (M.A.D.)
2   Department of Animal Nutrition, Sylhet Agricultural University, Sylhet 3100, Bangladesh
3   Department of Multimedia Engineering, Sunchon National University, Suncheon 57922, Korea
4   Interdisciplinary Program in IT-Bio Convergence System (BK21 Plus), Sunchon National University, Suncheon 57922, Korea
\*   Correspondence: yangcj@scnu.ac.kr; Tel.: +82-61-750-3235
†   Both authors have contributed equally to the manuscript as co-first authors.

**Abstract:** The size of the pork market is increasing globally to meet the demand for animal protein, resulting in greater farm size for swine and creating a great challenge to swine farmers and industry owners in monitoring the farm activities and the health and behavior of the herd of swine. In addition, the growth of swine production is resulting in a changing climate pattern along with the environment, animal welfare, and human health issues, such as antimicrobial resistance, zoonosis, etc. The profit of swine farms depends on the optimum growth and good health of swine, while modern farming practices can ensure healthy swine production. To solve these issues, a future strategy should be considered with information and communication technology (ICT)-based smart swine farming, considering auto-identification, remote monitoring, feeding behavior, animal rights/welfare, zoonotic diseases, nutrition and food quality, labor management, farm operations, etc., with a view to improving meat production from the swine industry. Presently, swine farming is not only focused on the development of infrastructure but is also occupied with the application of technological knowledge for designing feeding programs, monitoring health and welfare, and the reproduction of the herd. ICT-based smart technologies, including smart ear tags, smart sensors, the Internet of Things (IoT), deep learning, big data, and robotics systems, can take part directly in the operation of farm activities, and have been proven to be effective tools for collecting, processing, and analyzing data from farms. In this review, which considers the beneficial role of smart technologies in swine farming, we suggest that smart technologies should be applied in the swine industry. Thus, the future swine industry should be automated, considering sustainability and productivity.

**Keywords:** smart farming technology; sustainability; swine production

## 1. Introduction

In the swine industry, it is important and challenging to monitor the health and behavior of the full herd of swine. Therefore, smart tools are needed that can remotely monitor the pigs and can provide accurate information to the farm operator [1]. As the profit of swine farms relies on the normal health and behavior of swine, the industry owner demands the good health of pigs, ensuring optimum body weight, height, behavior, posture, and glossy skins, with the absence of any symptoms of diseases, malnutrition, or cataracts [1,2]. A remote monitoring tool can help to identify the health and behavior status of individual pigs, i.e., their temperature, posture, body weight, estrus, gait, abnormal behavior, etc. [1]. Good farming practices can ensure healthy animal production through detecting illness in animals early and can ensure economic benefits of the farm; meanwhile, poor farming management systems lead to disease incidence in farms, resulting in economic

losses to industry owners. Presently, the physical management and manual observation of pigs includes the operation of farm activities, such as feeding, cleaning, rearing, weighing, monitoring, etc., as well as identifying the health status of pigs, including illness symptoms, estrus, welfare, and the behaviors of pigs, etc. However, manual activities usually take a long time, with high labor costs, and frequently lead to operators missing the early identification of sick animals in the herd, with estrus resulting in poor performance from swine. Such an inferior farming system may lead to higher diagnosis and medicinal costs, resulting in economic losses for farm owners [3]. Considering the weak points of manual management, smart equipment—such as automated tools or sensors that can detect behavior and health characteristics as well as providing the information necessary for farms to operate their farming activities—might be useful in the modern pig industry [4].

Application of ICT knowledge in agriculture is not only useful for smart livestock farming but is also important for communication between government, farmers, and middlemen, associated with markets and consumers, etc.; meanwhile, the interaction may have a role in improving the production of the farm, as well as ensuring food security [5,6]. Therefore, a possible solution to this issue is presented in establishing sustainable livestock production through implementation of precision livestock farming, where smart equipment can be applied in farming systems with a view to increasing animal production with a modern management system. This is why sustainable livestock farming is very crucial in the modern era. Precision livestock farming refers to an automated management concept, focusing on remote detection, sound and image analysis, and observation of the body weight and behavior of individual animals, so that farmers can easily detect the health status of farm animals [7,8]. Present smart swine farming is a subcategory of precision livestock farming. The concept of smart livestock farming originated in the USA during the 1970s and 1980s, through the development of precision agriculture, based on global position signals (GPS). This technology has been applied to real-time data collection with precise location [9]. Later, precision livestock farming started in Europe in the 1980s and 1990s, based on satellite and radio frequency identification tags (RFID). The developmental stages of smart livestock farming consist of three generations: the first generation involves the network, sensor node, control node, and the optical monitor; the second generation focuses on environment control (housing environment), health management, and the big data cloud—IoT > cloud > AI analyzes the situation > decision/control by AI; finally, the third-generation model consists of energy management and the automation of farming [9,10].

However, the systems of environmentally monitoring the pig shed and the rearing of pigs with modern technology (smart farming), based on the Internet of Things (IoT), are not broadly applied to date. For example, since 2018, out of the total number of 23,000 farms, only 1425 livestock farms (about 6.2%) were run with smart livestock farming technology in Korea [9]. In this review, smart swine farming technology (auto identification, automated health monitoring, automated feeding, automated weighing, detection of behavior and welfare, remote monitoring, etc.) have been highlighted based on previous findings and future strategies. Thus, the target of this review is to deliver a general overview of, and guidance about, the tools and applications of smart equipment in the modern swine industry.

## 2. Importance of Smart Swine Farming

According to the report of the FAO and the UN, compared with the current population (7.3 billion) and the food production level, about 50–60% more food will be needed for to provide for the increase in the population, which is estimated to be 33% higher (9.7 billion) by the year 2050, while about 113 million people are currently suffering from food insecurity [11,12]. The grain and food self-sufficiency rates are about 24% and 53%, respectively, in Korea. This indicates that the Korean food security index is only 32, which is lower than other developed countries [9]. The future generation will require more choice of meat, milk, eggs as well as other livestock products which may lead to a significant food insufficiency, especially of food originated from livestock [13]. The economy of many developing countries mainly depends on agriculture (livestock sector), and the roles of

animals in producing a substantial amount of protein-rich food is encouraging livestock production [5]. With this concern, FAO highlighted the significant role of ICT-based smart agricultural farming as an effective way to satisfy the higher demand of food for future population [14]. Smart swine farming simply refers to the application of Information and Communication Technology (ICT) in the swine industry. As a developmental strategy for the swine industry, the ICT-based smart swine farming system is now a time demanding issue. The technologies, such as sensors, robots, remote monitoring, weather satellites, etc. have opened a door of innovation in the farming system [15]. The application of these innovative technologies could motivate the farmers, companies, and institutions to work together, increasing farm productivity [16]. These technologies are data based and the target data will be generated by internet systems [17]. New agricultural applications in smart swine farming and precision livestock farming through IoT, ICT, big data, and robotics will enable industry to increase the farm operational efficiency by lowering costs, reducing waste, and improving product quality. IoT technology based on wireless sensor networks is emerging quickly in agricultural farming, including livestock sectors [18]. During smart farming, all the parameters can be monitored, handled, or controlled by a software application or PIC (programable intelligent computers). IoT technology can also be used to influence swine farming by using real-time sensing, data analysis, information technology, and decision-making to improve swine health, welfare, and production efficiency.

In current years, swine farms have been rapidly changing the subtleties and competence of swine production patterns. The modern swine industry focuses on developing infrastructure and applying technology to the design of a feeding program, monitoring, health, welfare, and production performance [19]. Therefore, pig farmers strive to raise the profit of production and ensure food safety with the technological standards of pig farming [10]. In addition, the application of sensors (cameras, microphones, accelerometers, or radio-frequency identification transponders), the images, sounds, movements, and vital signs of animals are now the subject of scientific attention in modern swine farming to ensure the good health, welfare, and optimum production of pigs [10]. The productive performance of pigs, as an indicator of business profit, depends on various housing conditions [20]. Commercial pigs encounter many stress factors, including stocking densities, limited possibilities of movement, handling by the producer, unbalanced temperature, humidity, $CO_2$, $NH_3$ inside the shed, etc. Weaning is a critical stage of commercial pig farming. Nutritional, physiological, immunological, and psychological disruptions of nursery pigs due to poor management systems resulting in reduced feed intake, increased incidence of diarrhea, loss of bodyweight, and higher mortality [21]. In addition, the consequences of stress could reduce appetite, lower immunity of pigs with reduced weight gain, and increase reproductive disorders and diseases incidence in pigs [10]. The most important welfare issue in swine is neonatal mortality, resulting from injurious behavior such as tail biting, which can be controlled by ensuring standard monitoring in pig farms [22]. Ahmed et al. [23] described a disease-predicting method involving use of a ZigBee-based wireless network to monitor the movement of infected weaned piglets. Smart swine-farming technology is urgently needed to avoid unnecessary stress of pigs reared in conventional methods caused by handling while being fed, taken care of, and monitored.

Monitoring the indoor farm environment based on IoT technology is very convenient because the updating and maintenance costs related to this technology are low [24,25]. Moreover, the application of IoT technology in piglet sheds can help record the data (of feed intake, water intake, body weight, illness, movement, behavior, excretion of feces, vaccination, medical treatment, wind direction, rainfall, ventilation, etc.) accurately. Besides the technological advances in feeding and rearing, farmers impose close attention on the environment of the swine shed. Of all the parameters, temperature, humidity, $CO_2$, and $NH_3$ concentration are crucial for swine health. Modern monitoring technologies can provide scientific management to farmhouse owners that may help run farms effectively. In addition, the application of technological knowledge can provide business-related information such as marketing management, cost management, labor cost, disease manage-

ment, auto monitoring, analysis of biometric data, etc., to farm managers. A combination of these technologies may result in higher farm productivity. For example, pig farmers of Korea could improve the production rate by 5.7% (pig/sow/year) and 9.3% (market pigs/sow/year); and the feed efficiency has been increased by 9.3%, after the application of 1st generation livestock smart farming technology [9]. Thus the application of ICT knowledge in the swine industry would benefit the next generation.

## 3. Key Technologies Applied in Smart Swine Farming

Precision livestock farming tools or smart technologies can be fully functional in farms and can bring valuable returns to the farmers or industry owners [19]. In smart swine-farming technology, pigs are tagged with sensors that can measure animal responses as the primary indicator for individual physiology, performance, behavior, welfare, overall farm environment, etc. Moreover, through the application of the smart technology, farm owners can make appropriate decisions about daily activities, and receive early warnings about illness or any abnormalities happening in the farming system [19]. Through the technological implementation, data can be gathered automatically with in-house control computers or inside the farm computer. However, due to less familiarity with smart technology and intention to use those technologies, most pig farmers are missing the necessary information for operating the farm economically. Key technologies already applied in commercial swine production have been described below and some common smart tools are presented in Table 1.

**Table 1.** Applications of smart tools in modern swine industry.

| Smart Tool | Uses | Ref. |
| --- | --- | --- |
| Radio Frequency Identification (RFID) | Identification of pigs, individual animal data, date of birth, mortality, pen number, farm number or group number, etc. | Buller et al. [26]; Ariff et al. [27]; Maselyne et al. [28]. |
| Infrared thermal imaging | Temperature of individual pigs or whole herd, muscle injuries, infectious disease, ovulation etc. | Hristov et al. [29]; Rocha et al. [30]; Racewicz et al. [31]. |
| Microphone/cough detector/sound detector | Detection of normal or abnormal sound for sickness, coughing sound, heat detection, group behavior of pigs | Berckmans, [32]; Chung et al. [33]; Ferrari et al. [34]. |
| ZigBee technology | To detect environmental parameters: temperature, relative humidity, concentrations of carbon dioxide and ammonia in pig house | Zeng et al. [35]; Ahmed et al. [23]. |
| Deep learning/image analysis | Auto locomotion, movement pattern, behavior, posture, tail biting, temperature, body weight, etc. | Alameer et al. [36]; Nasirahmadi et al., [37]; Kashiha et al. [38]. |
| Two-dimensional (2D) cameras | Individual identification of pigs based on color, pigs' locomotion, behavior, posture, house environment, etc. | Riekert et al. [39]; Nasirahmadi et al. [40]. |
| Three-dimensional (3D) cameras | Feeding time, body weight, posture, lameness, injuries, group behavior, etc. | Death et al. [41]; Pezzuolo et al. [42]; Stavrakakis et al. [43]. |
| Accelerometer | Pigs' movement pattern including standing time, posture, monitoring, and welfare, etc. | Chapa et al. [44]; Benjamin and Yik, [2]. |

### 3.1. Uses of Sensors Technology on Smart Swine Farming

Application of biosensors for animal health monitoring is of growing interest among animal scientists and a number of sensors are being commercialized as they can effectively provide farming data, including timely alarm about ill animals or any infections in the herd [8]. Some wearable sensors are used to identify illness animals early and thus help prevent diseases over the herd, resulting in lower mortality of animals and preventing economic losses. In addition, through the sensors methods, farmers can make appropriate decisions with regard to culling diseased and affected animals on time to prevent further spread of diseases so that it can reduce the medication cost as well as can reduces antibiotics

uses in farm resulting lower the risk of antibiotic resistance in animals or human through consuming animal products. For an instant, among the methods to detect the presence of antibiotics in food or animal body, biosensors have been proved to be most effective as they work with transducing devices and can easily identify the element. Later, the identifying element makes a pairing with enzyme/substance or antigen/antibody receptor complex. Then the transducer can identify such pairing contact and forward a detectable signal in response to body functions that can be analyzed [8]. However, the application of biosensors in the animal industry has been limited until now because the function of those sensing elements depends on many factors, including their types, size, and properties with many environmental factors [45].

Sensor technologies are usually applied to determine animals' body temperature [46]; behavior and locomotion [47,48], stress factors or signs [49]; for detecting sound [34]; detecting pH [50]; or identifying disease [51] and viruses and other pathogens [52] in animal's body. The general farm-monitoring activities becomes easier and reliable to farm mangers by using biosensors technologies that are connected with hand phone or desk devices. Those devices can be set up under the skin of animals, or inside a body part that could provide meaningful information about the health and behavior of animals [8]. It is a great concern to ensure the good health of swine to produce safe meat for human consumption and to prevent zoonotic diseases, and biosensors technology can play a vital role in preventing such health hazards originating from animals or animal-based food. Thus, the smart swine-farming technology will meet the demand for safe food and ensure the sustainability of meat production from the swine industry. Another sensor=based technology, namely microfluidics, is becoming widely popular for detecting disease diagnosis of animals as well as food safety issues. Regular monitoring of subclinical ketosis in lactating sows is important for maintaining good health of breeding sows as well offspring of the herd. Previously, a human ketosis detector platform based on determining β- hydroxybutyrate (β-HBA) was applied in animals to detect the subclinical ketosis in animals. However, due to variation in blood grouping in animal species this technology was proved as non-effective. Therefore, highly specific sensor-based microfluidics proved effective and reliable for estimating (β-HBA) for ketosis in animal samples [53]. Thus, the invention of microfluidics techniques plays a vital role in the health management of farm animals. Recently several microfluidics biosensors with high sensitivity and specificity have been developed to rapidly detect β-HBA within limits of 0.05mM concentration [53,54].

With the goal of remotely monitoring the pigs and receiving early alerts about sick pigs, a sensor board was inserted on the left ear of individual pigs and it was found to be effective to provide necessary data regarding welfare and behavior (posture, gait, external temperature, locomotion, etc.,) to keep a physical record [1]. Pigs showed unnoticeable change during and after mounting the sensor boards on their ears. The duration for data collection was about 75 min and it also ensured real time, fully automated remote data collection from a distance between 15.25–18.3 m [1]. For laboratory analysis of the data, the authors considered a different sampling rate, for example five sample/sec to one sample/10 min, and the sleep time of sensors board while data are adequate. However, from this study the authors pointed out some drawbacks such as the developmental cost of electronic sensor boards, battery life, risk of damaging the sensor board, the size of sensors, etc. Therefore, larger ear tags and a reduction in the size of the sensor board would be beneficial. Furthermore, there are limits on different behavioral characteristics that machine intelligence can correctly identify. For instance, an individual herd's fixed machine learning system may not act appropriately on various farms, and some modifications may be needed [1]. Now functional base specific sensors have been developed for the farm demanding issue and the technological advantages. For example, environmental sensors are used to collect data on environmental parameters. A commercial humidity sensor (DHT11 sensor, Adafruit Industries, New York City, NY, USA) is used to detect the moisture in pig shed or farms. Similarly, indoor air flow transducers, gas sensors (for $CO_2$, $CH_4$, $H_2S$,

and NH3), infrared camera/3D camera and web cams (Pig CamTM) are used to collect the image of pig shed or individual pigs.

The output of those sensors such as sound- or image-analyzing technology can detect respiratory diseases or abnormal posture that indicate pigs' health, welfare issue or behavior. After receiving the excess value to sensor signals, it will provide an alert message or sound so that farm managers can take an appropriate action to prevent losses in farm production. Thus, it is an excellent opportunity to minimize the health problem before spreading it over the farm, which would be hard to control later. Collectively, the sensor system of monitoring is the most reliable because it could work on 24h/d and 7d/wk [19].

### 3.2. Smart Technology on Monitoring the Health of Pigs

Automatic detection of health is an essential practice in modern swine farming and now is a growing interest in research for detecting the health-related problems of swine. Racewicz et al. [10] reported that the automatic detection of health and behavior of swine would be the most effective and sustainable technique for optimum management of the large herd. Through the record monitoring system, a lot of data can be stored which would be useful for the next farming operation. However, the economic importance of auto health detection of farm animals is still under consideration. The factors affecting the auto health detection of pigs includes specific devices for effective measurement, technical knowledge of farmers, farming system, herd size etc., [55]. Animal health can also be measured by individual feeding or drinking behavior including feeding time, visiting number of feeding troughs or drinkers, etc. For example, lower feeding time with higher intervals of feed intake in a day may result from diseases or any abnormal physiology of pigs or abnormal environment in farms, which may create a problem for farm production [28,56]. The RFID system is capable to measure the feeding or drinking behavior including time of feed intake or water intake and visiting number of individual pigs to specific feeding area either nutritive visit or non-nutritive visit [57–59].

### 3.2.1. Pig Cough Monitoring

Real time monitoring of farm and health status of pigs can be achieved not only by video image analysis but also with sound analysis techniques. The sound produced by animals can play a role in detecting their physical needs such as hunger, sexual mating, signs of heat stress, and respiratory diseases such as cough or other illnesses [60,61]. Among the illnesses, respiratory diseases cause most discomfort to pigs and are responsible for changes in their vocal sounds such as coughing or sneezing [62] and the authors stated that a cough sound was also an indicator of indoor air quality, thanks to the invention of microphones that can differentiate the specialized sounds related to respiratory problems (infections) from the other cough sounds due to indoor dust or ammonia concentration or non-infectious cases [33]. Several studies have been conducted to detect respiratory problems of pigs based on sound analysis [33,34,63]. Ferrari et al. [34] showed that cough sound could be an effective indicator of respiratory infections such as pneumonia in pigs. The goal of the study was to compare the cough sounds caused by respiratory tract infection from inhalation of citric acid. The cough sound from pneumonia-infected pigs and from healthy individuals was considered based on various differentiate indicators, Root Mean Square (RMS), peak frequency, duration of coughing and time of a cough attack. The findings showed that RMS value and average peak frequency of cough sounds were significantly higher in non-infected pigs than in infected ones; but the length of the cough sound was higher in infected pigs than in non-infected groups. The authors concluded that their findings might have roles in the further development of real-time cough monitoring based on sound analysis, with a view to automatic monitoring and identification of infected pigs inside the farm. Another study by Exadaktylos et al. [63] stated that the accuracy rate of real-time detection of the illness in pigs by sound analysis was about 85%. Although many algorithms have been used to identify cough sounds of sick pigs, greater accuracy at the practical level still needs to be established. Considering the issue, Yin et al. [64]

proposed an algorithm based on the AlexNet model and structure of the spectrogram. In this proposed method, the sound signals will be converted into spectrogram images for identification and will show a higher accuracy rate. The authors compared their proposed algorithms with existing algorithms and found a higher accuracy rate of 96.8% and 95.4% on cough and overall recognition, respectively [64]. As an output of continuous research on sound analysis, now it is possible to identify the sick pigs 2-12 days before the conventional farm diagnosis by farmers or veterinarians [65]. In addition, a cough sound can be used as a biomarker for indoor air pollution of a farm [66]. However, the noisy environment inside the farm is the main drawback of sound-based health detection of farm animals.

### 3.2.2. Temperature Detection

The physical temperature of farm animals is an important parameter to ensure their health status and welfare. For example, various physiological changes including estrus, pregnancy, parturition, lactation, etc., can affect the body temperature [8]. Conventional techniques for taking body temperature based on core organs (heart, viscera, brain) and rectal passages have many limitations, including a high variation on temperature reading and risk of diseases transmission through the instruments/devices [8]. Some common tools, namely thermistors, thermocouples and infrared radiation sensors have been developed to measure the body temperature of farm animals [8]. Recently, microchips have been used to measure the peripheral temperature and those chips are inserted into the muscle/skin of pigs [8,67]. However, the presence of sub-cutaneous fat that can affect measuring the body temperature of animals; it showed a high variation in findings, thus this technique is now moderately reliable [67]. For example, because of subcutaneous fat the body surface temperature of the adult pig is lower than growing pigs/piglets [46]. Therefore, an image thermos-gram/thermos-vision based on infrared thermal image analyzing system for measuring body temperature at different environments has achieved the greater attention in modern swing farming [46,68]. The accuracy of this image-analyzing technique is for measuring temperatures up to 0.08 °C [68]. In addition, this technique is completely non-invasive and has no risk of spreading diseases [69]. Furthermore, the vulval temperature by infrared thermography can provide necessary information of ovulation in sows. A study by Simoes et al. [70] reported that vulvar–gluteal temperature was higher at pre-estrus and highest within 24–48 h of estrus, while the temperature declined at the lowest value after 6–12 h of estrus. Thus, the authors claimed that the vulval temperature could be a potential predictive marker for ovulation in sows. It has been hypothesized that higher levels of plasma estrogen may increase the vaginal blood flow resulting in the higher vulval–gluteal temperature of sows. In addition, greater blood circulation due to the higher level of estrogens in plasma was also responsible for redness and swelling of vulvar parts, which is known as an important signs of estrus in sows [71]. Another study by Skyles et al. [72] used a digital infrared thermal imaging technique to identify whether the gilts are in estrus or diestrus by measuring the vulval temperature. The authors found that the average vulval temperature was 35.6 °C at estrus, which was higher than diestrus (31.8 °C). The authors highlighted an important point that this technique could help farmers or researchers detect the silent heat of animals. In some previous studies, digital infrared thermography was successfully applied to detect the fertility status of bulls [73,74]; dairy cows [29]; and wild animals such as giraffes, pandas, and zebras [75,76]. However, the application of this technology in domestic farm animals is still in consideration. This is why the scientists prefer to apply Infrared thermography as potential tool to understand the physiological changes by measuring the temperature.

### 3.3. Smart Technology on Swine Farm Management Practice

The conventional method for keeping farm records or monitoring animals by recording data in text, or in a notebook or a device without data sharing, is inadequate for a highly profitable farm [8]. Day by day the pork market size is increasing due to consumer demand resulting in greater farm size for swine. Thus, providing the necessary care to the individual

pigs on time becomes critical for the farmers. The conventional farming system depends on land and labor creating a crisis of natural resources and animal welfare issue [77]. Therefore, modern swine farmers might need to use automatic tools to monitor swine health, welfare, behavior, and the farm environment [38,78]. In a first-generation model, Global Positioning Systems (GPS) were used. Unfortunately, obtaining data from GPS requires the whole field maps connected to a satellite and this was reported to be a costly process. Similarly, the limitation for a voice-entry system was also highlighted for its noisy backgrounds inside farm. Recently, a radio-frequency identification (RFID) technique has been applied in smart swine farming where the RFID tags can be attached or inserted into the animal's body to receive the health and behavior data from an individual or from the whole herd [79]. The RFID and accelerometer techniques are now well incorporated; however, other smart technologies are also gaining higher attention in modern pig farming [26]. The development of precision livestock farming technology, which refers to auto identification, an auto weight-detector device, an auto water meter, an auto feeder, an auto temperature-detector device, remote monitoring for behavior, pig cough detection, and a camera system for visual data collection, is now commercially known, but the business operating management software is still in investigation [19]. With the help of IoT technology, a device will be connected to the internet, resulting in easy decision making, and the farm can be operated automatically or remotely [80]. Recently, the application of a robotic system for cleaning, and sanitizing the floor or housing materials, is a great invention of modern technology to save time, labor and prevent zoonotic diseases originated from farmhouse or farm products. A robotic machine is an automatic high-pressure device that can move around and perform farm operations in pig houses [81]. Similarly, the tasks (electric or $CO_2$ stunning systems, carcass washing, scalding and dehairing, cutting parts, etc.) in a slaughterhouse can be automated by machinery or a robotic system to ensure the carcass is free from contamination, saving labor, time, money, etc. [81]. However, further development of a robotic system is needed to make it smarter to perform its function so that it would be applicable in commercial farms with an affordable cost. In addition, the smart technologies (RFID, bar code, tags etc.) applied when packaging the farm products are also useful to obtain the information of shelf life, expiration date, preservation temp, and quality of product in smarter ways.

### 3.3.1. Auto Identification of Pigs

Identifying the individual pigs from a group of pigs or pen is very important for specific care, medication, separation or sorting from group, or to keep uniformity. Electronic devices based on touching, image processing, code reorganization, scanning, and trans-receivers are now used to identify the individual animals as the benefits of precision livestock farming [19]. In addition, to identify a particular pig for farm management or research purposes, various tools such as RFID and facial recognition systems are becoming popular in the swine industry [2]. Many studies reported that an RFID chip is the most effective tool currently in practice to recognize pigs in modern swine farms [82,83]. The device is inserted on the ears and the chip can store the information. Different radio frequency waves act as a communication medium between tags and RFID chip readers. The device can generate signals which transfer the data to the readers within the range. Finally, the data can be saved and analyzed, and the RFID chip is used to identify the animals [27,83]. Furthermore, low-frequency RFID techniques were also applied for electronic sow feeders in group pens, visiting time of feeder by each sow, time limit for feeding, as well as identification of pigs [59]. However, low-frequency RDIF has limitations including lower reading range and inability to recognize more than one animal together [84,85]. Therefore, ultra-high-frequency readers are now used in modern swine farms to identify group animals with a greater range of about 3 to 10 m. However, UHF readers are sensitive and can include stress, pain, injuries etc., as they are attached tightly into the ears of swine. Other demerits of RFID are losing the tags and removing the tags before slaughtering. Considering the adverse effects of RFID, Alameer et al. [36] reported that video imaging

that involves 2D or 3D camera-based deep learning can be an effective alternative to RFID for detecting the feeding behavior of pigs, since it is cheap and simple to use and needs no sensors or markers. Furthermore, the deep learning method could auto separate the non-nutritive visit of pigs to the feeding area from the actual feeding with an accuracy rate about 99.4% [36]. Therefore, further research is needed to adapt these technologies to auto-identification of individual pigs and their feeding behavior.

### 3.3.2. Automatic Weight Detection

Keeping the optimum body weight of pigs is an important parameter to make a herd economically successful [86]. Many stressors and other factors can cause lower feed intake, resulting in lower weight gain in pigs. Thus, regular recording or monitoring the growth rate of pigs is very important to ensure the sound health of the pigs, as well as for a highly profitable swine farm. Obtaining accurate data about weight gain and the uses of these body weight data to make appropriate decisions for marketing or slaughtering fattening pigs is essential in smart swine production [86]. As a part of precision livestock farming, automated weight monitoring of pigs can be performed using image analysis as a smart tool in modern pig farming. Kashiha et al. [86] reported that the accuracy rate of fattening pigs' weight was about 97.5% (0.82 kg errors) at group level and 96.2% at an individual level (1.23 kg errors) through image analysis on a real-time basis. The findings were appropriate since the existing method for weight detection was about 95% with 2 kg errors in automated tools and 97% with 1 kg error in passing through a method by a corridor system. In this study, the authors used grower pigs with four pens for ten pigs, each pen having an average initial body weight of 23 kg and 45 kg at the ending period of the experiment. In this method, as a shape-recognizing technique, unique painting was applied to separate individual pigs and ellipse-fitting processes were used to confine pigs in a fixed image. Later, the specific part of the pig inhabiting the ellipse was considered, and the final body weight of the pig was then calculated based on the dynamic modeling. The model was validated by matching the actual manual weight of the individual pigs. Thus, this imaging technique can provide accurate automated body weight of individual pigs whereas the handling of pigs frequently by manual methods causes heavy stress with a negative effect on production and is labor consuming [86,87]. Besides, smart techniques for weighing pigs can help to identify the slow-growing pigs and farmers can take appropriate decision to manage this effectively. Some studies noted that the body weight of pigs can be estimated by using the image of the top view of the body area [88]. Many commercially available auto-weighing smart tools have been developed based on image-analyzing technology, namely, eYeScan by Fancom BV, Pigwei by Ymaging, OptiScan, GroStat, and WUGGL, etc. These available techniques can provide weight value with average errors around 1.5 kg. The device connected to a camera contains image-processing software that generates weigh data for a group of pigs or individual pigs [19]. Lee et al. [89] analyzed and verified machine learning technology (MLT) for estimation of growth performance data including daily weigh gain, feed intake, and market pigs per sow per year considering the environmental factors such as temperature, humidity, and pigs' initial body weight, stocking density, etc. Data were collected from 55 pigs in South Korea for a period of one year. The findings noted that MLT showed about 28% higher precision rate for growth performance data and the temperature was the most affecting factor for body weight gain of swine. The authors concluded that MLT-based prediction is the most effective way to estimate the growth performance of pigs.

### 3.3.3. Remotely Monitoring Behavior (Feeding) and Welfare of Pigs

Through the remote monitoring system, the swine farm owners can manage multiple sheds with minimum time and can obtain data from each shed or from individual pigs. The remote monitoring systems will provide up-to-date information to the farm manager which can also ensure good health, welfare, and enhance the performance of pigs [19]. Among the smart monitoring tools, video monitoring is the most popular technique practiced

in modern swine farms. With the help of a video monitoring system, the behavior of a group of pigs or even the behavior of individual pig can be detected easily [2]. Further, it can provide data about swine movement including location and position by analyzing the image data [90] with activity such as movement [38]. Imaging techniques are also used to measure the body weight [42] of pigs, lameness [43] of breeding pigs, feeding behavior [91], aggressive behavior [92,93], and estrus or heat [94] of pigs. For a few years, two-dimensional/2D monochrome and color camera devices have been used as smart video tools considering cost effectiveness and their functional role in farms [10]. However, several scientists have suggested various methods to estimate the body size or body condition based on images analyzed from a 2D camera [91]. In addition, Kashiha et al. [86] successfully estimated the individual pig weight by 2D image analysis. The result may vary due to the location of camera and the height of camera position from animals. Besides, animal density may affect monitoring, image analysis and weigh estimation [95]. Circadian rhythms monitoring systems based on real-time video capturing have been developed [96]. This technology can detect the dust concentration in a swine pen with its response to the behavior of pigs. In addition, various image processing systems can detect the thermal comfort zone inside the pig house by analyzing the pigs' behavior [97]. Besides machine learning techniques have been applied to monitor swine farms [89,98]. For example, the sickness sign or abnormal behavior of swine can be detected by a deep neutral network [99], and the amount of water consumption can be indicated by different MLT [100].

One of the greatest advantages of the deep learning method is to detect the feeding behavior of swine, including the feeding time which has value for nutritional studies [101] as well as health and welfare [91]. Sudden changes in feed intake or feeding time by pigs can ensure the environmental stress or health issue [95] and changing the feeding pattern may ensure sudden outbreaks of diseases [56]. Therefore, analysis of feeding behavior to estimate the feeding time of pigs is useful to evaluate the performance of individual pigs. Lao et al. [102] have developed a method for detecting feeding behavior based on video image analysis of body parts (head and shoulder) and feeder. According to their study, sows were considered to be engaged in eating if the height of the head seemed to be smaller than the feed trough; the space between the head and shoulder was shorter than that of standing sows. Besides, a depth-sensor-based image analysis method was used to evaluate the feeding behavior of sow considering the part of the sow's head seized within a pre-defined portion of the feed trough [103]. There were several studies in which the feeding behavior of pigs was considered based on putting down the head [104], putting the head in the food area, biting, smelling and chewing of food, and grubbing with snout inside feeder trough [59,102]. However, if the sows were controlled in farrowing stalls, the dimensional relationship between the sow's head and feeder is tiny to recognize the feeding behavior of an individual sow. Therefore, to identify the feeding behavior of individual pigs from a group of sows, such as to identify of sows from its piglets, a method—namely a faster region convolution neutral network—was developed which was based on the ratio of the sow's head to the feeding region [105]. In order to provide a guideline for recognizing the daily behavior of sows based on image analyzing techniques, a total 468,000 frames from three sows were considered during period of 26 h [105]. The authors noted the behavioral classification, which included i) drinking (97.49%); ii) feeding (95.36%); and iii) nursing (88.09%). Furthermore, the time distribution on behavioral pattern included inactive behaviors (69.34%); nursing behavior (14.50%); medium activities (8.38%); feeding (4.04%); drinking (2.26%); and moving activities (1.48%). Finally, the authors concluded that the proposed method is very useful for automatic recognition of sow behaviors from image analysis, which will help the pig farmers operate the farm activities. Compared with traditional management, smart swine-farming equipment such as automated monitoring tools, which can be controlled from outside by IoT technology so called deep learning or image analyzing, might be a potential strategy for modern pig farming.

However, the major limitation of practical applications of those smart technologies in swine farming includes high installation and maintenance fees, difficulties in using the new technologies due to lack of knowledge or skill of farmers, lack of confidence in the manufacturing company or technologies, etc. Therefore, to adapt the smart technologies widely, manufacturers have to focus on low production costs to develop smart equipment or technology and ensure the quality of the equipment to establish reliability in the market. Training opportunities to farmers, post-service of equipment, and routine feedback of technological applications in farms need to be considered.

## 4. Conclusions

Although recent scientific studies have highlighted the effectiveness of using smart tools in modern swine farming, the IoT-based rearing and monitoring system for pigs' shed environment is not broadly used. Many technologies still need to be adapted in the modern pig industry, which is the target for the third-generation model or green revolution. Therefore, to overcome the challenge of modern pig production considering pig health, environmental stress, auto monitoring, behavior and welfare issues, as well as to modernize the swine farming industry and make pig production sustainable, technological innovation urgently needs to be imposed. This review study provided a basic idea of remote monitoring (auto identification, auto feeding, auto weighting, auto health monitoring, etc.) of pig farms with smart technology. Data on environmental parameters and husbandry methods with smart rearing, feeding, and monitoring for large-scale pig farming will meet the practical demands of pig production, which would benefit the next generation. Thus, the future development of smart swine farming is to be highly expected.

**Author Contributions:** Conceptualization, C.-J.Y.; writing original daft preparation, S.M.; preparation of manuscript, S.M., H.-S.M. and M.A.D.; editing and revision, S.M. and C.-J.Y.; supervision, C.-J.Y. All authors have read and agreed to the published version of the manuscript.

**Funding:** This work received no external funding.

**Institutional Review Board Statement:** Not applicable.

**Informed Consent Statement:** Not applicable.

**Data Availability Statement:** Not applicable.

**Acknowledgments:** The author would like to express their sincere gratitude to the Brain Pool Program supported by the National Research Foundation of Korea (NRF) and Department of Animal Science and Technology, Sunchon National University (SCNU), Korea for his Post-Doctoral study under Chul Ju Yang, grant number 2021H1D3A2A01099735. The authors also would like to acknowledge the Korea Institute of Planning and Evaluation for Technology in Food, Agriculture, and Forestry (IPET) and the Korea Smart Farm R&D Foundation through the Smart Farm Innovation Technology Development Program, funded by the Ministry of Agriculture, Food, and Rural Affairs (MAFRA) and Ministry of Science and ICT (MSIT) and Rural Development Administration (RDA) (421023-04).

**Conflicts of Interest:** The authors declare no conflict of interest.

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
