# Peer review of "Applications of Smart Technology as a Sustainable Strategy in Modern Swine Farming"

_sustainability, doi:10.3390/su14052607_

Round 1
Reviewer 1 Report
1,the abstract needs to be shortened to concisely include the major ideas and structures of the whole review.
2, the introduction part should include less irrelevant background, and focus more on the meaning to choose this topic and the objective of this review work.
3, part 1 should more focus on concepts of smart technology, which is the core concepts according to your title. For the importance, that should be included in the introduction part.
4, it seems sensor technologies were mainly reviewed in part 2, 3 and 4, and some contents were reviewed in different parts, e.g. 2D and 3D cameras. Therefore, the relevant summary should focus more on the “smart sensor technologies in swine farm”, and the repeated contents should be combined and re-organized.
Author Response
Reviewer 1
Authors response (AR): Dear Reviewer, thank you very much for reviewing our manuscript with good necessary comments, that’s all have been highly accepted by us. We have revised our manuscript accordingly. Below the point to point responses for your kind consideration.
- the abstract needs to be shortened to concisely include the major ideas and structures of the whole review.
Authors response (AR): Dear Reviewer, Thank you very much for your good comments and valuable time for us. As per advises The abstract part has been shortened by deleting L-30-35 as less important information. Please check the abstract part L15-32; Thank you.
2, the introduction part should include less irrelevant background, and focus more on the meaning to choose this topic and the objective of this review work.
Authors response (AR): Thank you very much for your good comments. As per your suggestion, we have revised The Introduction part, more concise with necessary information and by deleting less relevant statement. (deleted L-62-64; 65-77 as less important information for Introduction part). We have replaced that information under the section 2 (L89-101; L135-137). Thank you.
3, part 1 should more focus on concepts of smart technology, which is the core concepts according to your title. For the importance, that should be included in the introduction part.
Authors response (AR): Thank you very much for your good advises. As per your suggestion, we have replaced the information related with concept of smart farming in Introduction part (L- 69-77). Thank you.
4, it seems sensor technologies were mainly reviewed in part 2, 3 and 4, and some contents were reviewed in different parts, e.g. 2D and 3D cameras. Therefore, the relevant summary should focus more on the “smart sensor technologies in swine farm”, and the repeated contents should be combined and re-organized.
Authors response (AR): Dear reviewer thank you very much for your good comments. As you know most of the smart tools based on sensors, so we have focused it in a full heading (3.1), And rest of part we have designed on the farming operations/ activities by using smart tools/sensors. However, as per your suggestion we have changes the subheading number to make a good connection with the main heading under 3 (Key technologies….). Also we highlighted the major sensors base tools with its application under 3.1 (Uses of sensors technology on smart swine farming) as per your suggestions.
Hope you will consider our revised submission with your kindness and thanks a lot for your valuable time for us.
Thank you.
Reviewer 2 Report
The proposal presented is suitable for the magazine, excellent structure and timely and updated information, with great impact in the area, I only recommend modifying the quality of the image proposed in the article and if it is not relevant, eliminate it since I don't see what it contributes to the document.
Author Response
Reviewer 2
The proposal presented is suitable for the magazine, excellent structure and timely and updated information, with great impact in the area, I only recommend modifying the quality of the image proposed in the article and if it is not relevant, eliminate it since I don't see what it contributes to the document.
Authors response (AR): Dear Reviewer, Thank you very much for your good comments and valuable time for us. As per advises the images have been removed in this revised submission as all the necessary information about smart tools were provided in text. Again thank you very much for your time and suggestions. Thank you.